# A Financial Management Platform Based on the Integration of Blockchain and Supply Chain

He Liu [1], Biao Yang [1], Xuanrui Xiong [1,*], Shuaiqi Zhu [2], Boyu Chen [1], Amr Tolba [3] and Xingguo Zhang [4]

1   School of Communication and Information Engineering, Chongqing University of Posts and Telecommunications, Chongqing 400065, China
2   School of Software, Dalian University of Technology, Dalian 116024, China
3   Department of Computer Science, Community College, King Saud University, Riyadh 11437, Saudi Arabia
4   Department of Mechanical Systems Engineering, Tokyo University of Agriculture and Technology, Nakacho Koganei, Tokyo 184-8588, Japan
*   Correspondence: xiongxr@cqupt.edu.cn; Tel.: +86-13983272758

**Abstract:** Internet of Things (IoT) finance extends financial services to the whole physical commodity society with the help of IoT technology to realize financial automation and intelligence. However, the security of IoT finance still needs to be improved. Blockchain has the characteristics of decentralization, immutability, faster settlement, etc., and has been gradually applied to the field of IoT finance. Blockchain is also considered to be an effective way to resolve the problems of the traditional supply chain finance industry, such as the inability to transmit core enterprise credit, the failure of full-chain business information connections and the difficulty of clearing and settlement. Supply chain finance allows the strongest enterprise in the supply chain to apply for credit guarantee from the bank to obtain bank loans, and use the funds for circulation in the supply chain to ensure that each enterprise in the whole supply chain can obtain working capital to realize profits, so as to maximize common interests. In this paper, a financial management platform based on the integration of blockchain and supply chain has been designed and implemented. Blockchain is used to integrate supply chain finance to synchronize the bank account payment system, realize the automatic flow of funds, process supervision and automatically settle account periods based on smart contracts. The four functional modules of the system are designed using unified modeling language (UML), and the model view controller (MVC) architecture is selected as the main architecture of the system. The results of the system test show that the proposed platform can effectively improve the system security, and can use the information in the blockchain to provide multi-level financing services for enterprises in supply chain finance.

**Keywords:** blockchain; supply chain; financial management; Internet of Things finance



## 1. Introduction

With the developments in economy, human society and communication technology, the world has entered the era of Internet of Things (IoT). Many IoT applications have been developed and are changing people's lifestyles. IoT finance relates to different kinds of IoT applications which can expand financial services to the entire IoT commodity transaction, subvert traditional finance and internet finance services and make financial business processes more intelligent, transparent and accurate. Due to the varied characteristics of the IoT, the performance of centralized transaction management centers is relatively poor. Collecting all transaction information to the central server will cause incredible communication overhead, resulting in transaction delays and invalid operations [1]. In addition, a single point of failure can easily affect centralized operations, and various man-in-the-middle attacks constantly occur due to the financial value of transaction information. Blockchain, an emerging distributed ledger technology, is decentralized and can better meet the needs of the IoT. This has focused more and more academic and industrial research efforts onto

the blockchain and the IoT. Blockchain is rapidly evolving as a new basic tool for building powerful distributed applications. Similarly, the IoT is increasingly being deployed in areas such as smart cities [2], smart homes [3], smart healthcare [4], etc. The significant growth in IoT devices and the combination of blockchain technology and other applications [5] have brought opportunities for change to many industries. Researchers have recently developed a variety of application systems, e.g., blockchain-based electrical fault detection and safety maintenance transmission schemes in remote power grids [6] and the blockchain model for secure transactions of mobile commerce [7].

Blockchain technology has outstanding immutable, distributed, secure and decentralized features [8]. Therefore, blockchain has become the most popular technology in the field of internet finance [9–11], and it can be used to establish an internet-based supply chain finance platform to provide information and networks for financial chain businesses. Network-based supply chain business management is realized with the help of information processing technology, which could greatly improve the efficiency of supply chain-related business management [12].

In this paper, a financial management platform that combines blockchain and supply chain is designed and developed to support and improve enterprise business processes. For the core enterprises in the supply chain, we creatively integrate blockchain technology into the financial management platform. The technical features of the blockchain, such as credit transferability, information sharing, data traceability and tamper resistance, are cleverly used to solve the pain points of the traditional supply chain financial industry, such as the inability to transfer the credit of core enterprises, the inability to connect the business information of the entire chain and the cumbersome clearing and settlement.

The rest of this paper is organized as follows. Related work is presented in Section 2. The system design is introduced in Section 3. System implementation is described in Section 4. Section 5 covers system testing and discussion and Section 6 concludes this paper.

## 2. Related Work

This section first reviews some previous work on distributed processing and mobile networks and then presents some blockchain-related work to better understand our approach.

The maturation of 5G communication technology has promoted a new round of industrial revolution and supported the high-quality development of the economy and society [13–15]. Distributed technology has also been applied to different fields. In order to meet the requirements of an ultra-low delay for real-time traffic management in smart cities, Ning et al. [16] expanded the cloud computing function from the central network to the edge network to achieve distributed traffic management, thereby minimizing the response time for vehicle collection and reporting of the entire city event. Wang et al. [17] put forward a fog-enabled real-time traffic management system that enabled minimizing the average response time on the ground of vehicle-reported events and built a distributed urban traffic management system that integrates fog computing with the Internet of Vehicles (IoV) systems to provide computing resources to end users and ensure low latency.

The fast development of the IoT has led to the rising demand for ubiquitous connectivity and ultra-low latency [18], and the limited spectrum resources are becoming increasingly unable to meet the growing service demands of users [19,20]. There are many studies based on mobile networks to solve the computational resource allocation problem. For example, a mobile blockchain framework based on mobile edge computing (MEC) was proposed to protect the data and privacy security of mobile devices at the time of transactions in the industrial IoT. A corporate bandwidth and computational resource allocation problem was first formulated to maximize the long-term utility of all mobile devices, taking into account blockchain throughput and device mobility [21]. Driven by the growing demand for real-time mobile application processing, multi-access edge computing is considered as a promising paradigm for forwarding computational resources to the network edge [22]. The latest developments in content caching and edge computing in

wireless networks enable intelligent transportation systems (ITS) to provide high-quality services to vehicles. However, the wide variety in time-varying network states and vehicle applications make it challenging to efficiently allocate resources to intelligent transportation systems. Artificial intelligence algorithms possess the ability to recognize the time-varying and diverse characteristics of connected vehicle networks, enabling ITS intent-based networks to meet these challenges [23]. Vehicular traffic is an important part of modern cities. However, more and more traffic accidents and traffic congestion have become obstacles to the realization of smart cities. The increase in the asymmetry of traffic flow and the wide vehicle distribution make it essential for network operators to design intelligent offloading strategies to provide high-quality services and improve network performance to users [24]. While modern transportation systems facilitate the daily lives of citizens, air pollution and increasing energy consumption pose a challenge to the establishment of green cities. Ning et al. constructed an MEC-based energy-efficient scheduling framework for the IoV to meet the heterogeneous requirements of communication, computation and storage within the IoV, which minimized the energy consumption of roadside unit nodes under task delay constraints [25].

The combination of blockchain technology and intelligent transportation systems [26] or personal health records [27] has solved various security and privacy issues and personal privacy issues when sharing personal cases. The innovative combination of blockchain technology and multidisciplinary research provides opportunities for change in many traditional industries [28–30]. The internet-based supply chain platform also helps different enterprises in different regions to interact with each other through the network platform, which greatly reduces the management cost and cross-regional management difficulty. From a macro perspective, the establishment of a supply chain finance platform helps to integrate and allocate resources, which has a positive effect by improving economic and social benefits [31]. With the support of the supply chain finance platform, the credit behavior of marketing companies in the supply chain is effectively curbed, which plays an important role in improving the capital turnover rate and reducing the business risks of enterprises. After capital turnover is guaranteed, enterprises can invest more energy in product development and sales channel expansion, which is of great significance to improving their market competitiveness and economic benefits.

## 3. System Design

### 3.1. System Requirements Analysis

The internet-based supply chain platform also helps different enterprises in different regions to interact with each other, which greatly reduces the management cost and cross-regional management difficulties. In the long run, the realization of the supply chain finance platform [32] will have a positive effect on the integration and allocation of resources and the improvement in economic and social benefits. Based on the above needs, a supply chain finance management platform has been realized, which mainly includes four functions: registration and account opening, credential management, online financing and capital management. To ensure the security of the financial management platform, all registered enterprises on the platform need to undergo real-name authentication first, and also need to undergo real-name verification through CFCA's U-KEY and digital certificates each time they log on to the platform, to verify the real identity of participating enterprises and to carry out online confirmation and signature verification. The voucher management module is mainly used for voucher registration, voucher status management and voucher sign-off and transfer. The online financing module mainly includes three major functions: financing application, financing approval, view and financing repayment. After receiving the vouchers, suppliers can select various funders that have been stationed on the platform to initiate financing applications based on their own needs. After receiving applications for financing from suppliers at all levels, the funders will review the applications based on their credentials and other information, and the review results can be released online in real-time, making the whole process simple, efficient, accurate and convenient. Funds management is

the core function of the whole financial management platform, which is the integration of blockchain technology and supply chain finance technology. It relies on the decentralized, tamper-evident and smart contract features of blockchain [33], synchronizes the bank account payment system and realizes the automatic flow of funds, process supervision and automatic settlement at maturity. The entire fund management function mainly includes account management, fund clearing and top-up and withdrawal. The design of the overall functional structure of the financial management platform is shown in Figure 1.

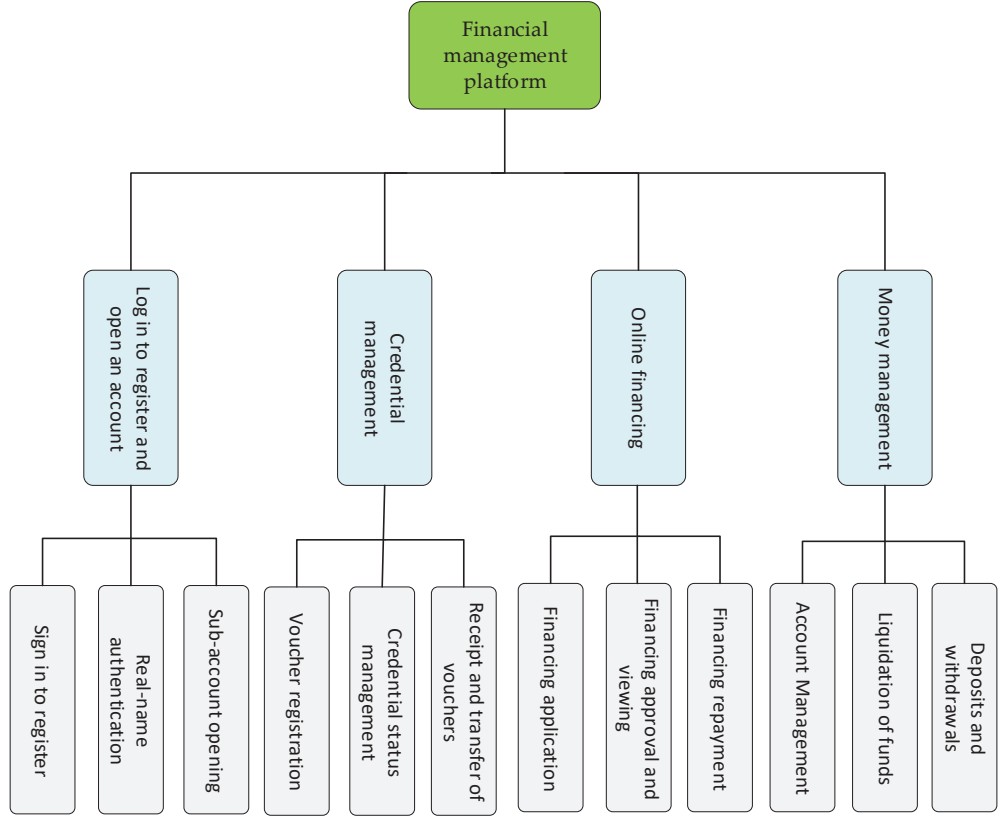

**Figure 1.** General diagram of system function structure.

To ensure that all functions of the financial management platform can function properly and meet the design requirements and standards as a prerequisite, it is not only required to analyze the system functional requirements but also analyze in detail the non-functional requirements such as system reliability [34] and security.

### 3.1.1. Reliability

In order to ensure that the developed system can operate stably and reliably in the financial management platform, it is necessary to ensure that the probability of failure of the system within the specified continuous operation time is less than the threshold to avoid frequent failures in practice that lead to system unavailability. On the one hand, to ensure the reliability of the system, the system also needs to be configured with a complete automatic error correction mechanism. As the subject development system stores sensitive information such as daily financial operation data of the supply chain in the actual application, to ensure the security of the system data, the system needs to take corresponding security measures during the database design. In addition, once the system detects an error in the system operation, the guidance mechanism should be activated immediately to guide the user to complete the required operation correctly.

### 3.1.2. Security

The financial management platform designed and developed in this topic will store a large amount of sensitive information such as financial data and company operation data of each company in the supply chain in practical application [35]. To provide comprehensive and effective security for the entire financial management platform dataset, in the process of system design and development, it is necessary to specifically configure a user identification and permission control module and then identify and recognize the corresponding user identity in the system. After the identity of all users of the entire financial management platform has been authenticated by the system, the system can set the operation permissions required by the users regarding their roles. After setting the corresponding operation rights, users can only operate and access the data with the operation rights they have. The user permission control mechanism can guarantee the security of the system data efficiently and reliably, and if the user has an abnormal operation, the system will immediately stop the user from accessing and responding to the corresponding operation.

Considering that sensitive information such as user passwords, system configuration parameters and enterprise fund data are stored in the system database, the system encrypts the sensitive data in the database to further improve the security of the system. In addition, data backup and disaster recovery modules need to be configured so that the system database can be backed up regularly and the platform can start the corresponding disaster recovery mechanism in the event of the destruction of the main service. At the same time, the backup data can be quickly restored to ensure the normal and reliable operation of the platform, thus mitigating or even avoiding the loss caused by the destruction of the platform.

### 3.2. *Design of the Overall System Architecture*

Through a functional and non-functional requirements analysis of the entire financial management platform and a detailed analysis and comparison of the main information technologies, the platform was mainly designed with MVC architecture to ensure the portability and scalability of the financial management platform. The user display interface is used to show the core business processes and the business support is used to ensure the normal operation of the system backend. The display layer and the business processing layer form the entire system framework. The data layer is used to ensure the security of data and the recall of system data. The detailed design of the overall architecture of the specific financial management platform is shown in Figure 2.

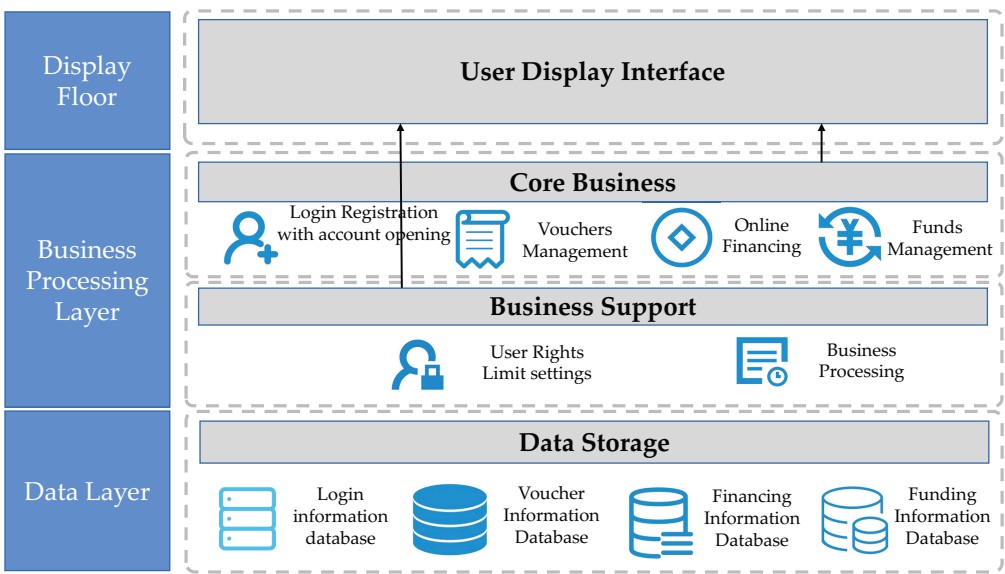

**Figure 2.** System overall architecture diagram.

From the overall system architecture diagram, we can see that the overall architecture of the financial management platform developed in this project mainly adopts the MVC architecture design approach, which can be divided into three levels: the presentation layer, the business processing layer and the data layer.

Presentation layer: The main function of using this presentation layer is to enable users to access the business operation level, and the platform uses the foreground to be able to visually display the business operations and system information in this layer, to provide specific docking services and operations for the business operations required by the users.

Business Processing Layer: This layer is the core layer of the financial management platform, which can provide support and business processing operations for the four main functions of the entire financial management platform: registration and account opening, credential management, online financing and fund management.

Data Layer: This layer enables the flexible retrieval and storage of data information in the information table of the whole financial management platform, for which the system mainly handles registration information, credential information, financing information and fund information.

MVC hierarchical architecture can easily complete distributed deployment: the bottom layer is the presentation layer, which is mainly responsible for the display of the interface directly facing the user; the second layer is the application system service layer, which is usually deployed in one or more servers and is mainly responsible for completing the logical processing of all kinds of data in the system; and the third layer is the data layer, which is mainly composed of the database system and mainly responsible for the management of all kinds of data in the system.

In terms of system network topology, the financial management platform network consists of a backbone network, a local area network (LAN) and a wide area network (WAN). The WAN enters the LAN through routers and firewalls for security screening to guarantee the security of the system platform. In most cases, staff access and operate the system from within the local area network. To meet some special cases where users cannot log in to the system within the LAN, the specific design provides a remote VPN login service to the system. The network topology of the financial management platform is shown in Figure 3.

The whole network of the system consists of two parts, i.e., an intranet and an extranet. The platform is able to use the extranet to achieve interaction and handle the account information of internal customers, and the enterprises can also use the network to achieve interaction and obtain customer account information. When staff need to access the system through the external network, they need to log in to the system through a VPN account and password. The security and reliability of the system are ensured through VPN accounts, which in turn facilitate the processing of key business information such as online financing and fund management.

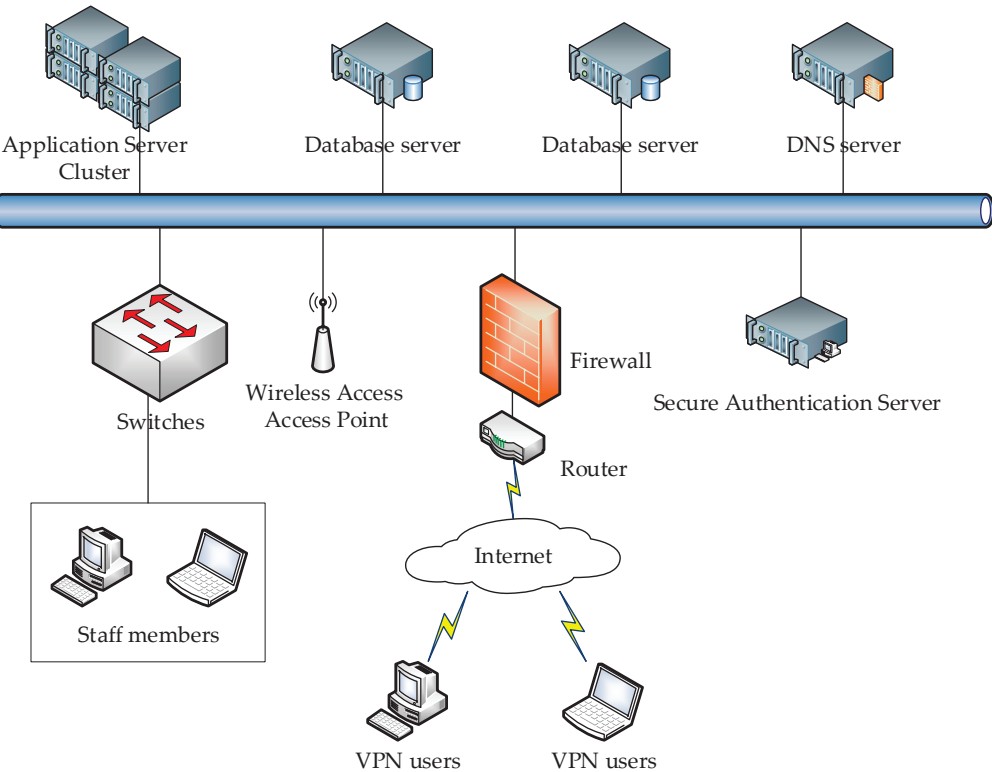

**Figure 3.** System network topology diagram.

*3.3. Detailed Design of System Functional Modules*

3.3.1. Detailed Design of Login Registration and Account Opening Module

According to the system requirements analysis, the login and account opening module are mainly used to realize the registration and account opening of all users of the platform and other related operations, while facilitating the opening of accounts by relevant corporate users and funders. The whole login registration and account opening module mainly includes three major sub-functions: login registration, real name authentication and sub-account opening.

The registration sub-module mainly includes two sub-functions: invitation code registration and registration information editing. When registering, enterprise users need to click "Register" on the login page to register on the platform, enter the invitation code received, then edit the required registration information according to the template requirements. After editing is complete, you can directly register and finally fill in the username and password, and use the just registered username and password to log in to the platform. According to the login and registration workflow diagram, it can be seen that the platform users need to edit the basic registration information in the login and registration operation, and then choose the way to receive the invitation code, and enter the invitation code after the user registration operation.

The real-name authentication sub-module consists of three core functions: real-name authentication information entry, information verification and payment password setting. All registered enterprises in the financial management platform need to undergo real-name authentication first, and they also need to undergo real-name verification through CFCA's U-KEY and digital certificate [36] each time they log in to the platform to verify the real identity of participating enterprises and to carry out online confirmation and signature verification. U-KEY and electronic signature use the CFCA digital authentication method. CFCA digital certificate is a piece of electronic data containing user identity information, user public key information and the CFCA digital signature. The digital certificate is the identity proof for all kinds of terminal entities and allows users to conduct information exchange and business activities online, ensuring the authenticity of the identity of both

parties to online transactions, the integrity and confidentiality of information and the non-repudiation of transactions. Only if the participating subjects are authentic can we ensure the authenticity of subsequent operations on the platform.

The sub-account account opening sub-module mainly includes two sub-functions: account opening information entry and account opening confirmation. In the process of account opening information entry, the main user needs to input the sub-account opening information, including SMS notification cell phone number, contact address, E-mail information, etc., which is to be entered directly after the account opening confirmation.

### 3.3.2. Detailed Design of Voucher Management Module

Based on the specific demand analysis of the voucher management function, the voucher management module is mainly used to realize all the high-quality enterprises of the platform to convert credit into collection vouchers that can be used in multi-level supply flow, splitting and financing. The entire voucher management module consists of three core sub-functions: voucher registration, voucher status management and voucher sign-off and transfer.

The voucher registration sub-module consists of two core functions, voucher registration information entry and voucher batch registration. Voucher registration can be done by inviting upstream suppliers with real names to issue vouchers, filling in the signatory and payment-related information, choosing whether to upload accompanying trade information, clicking "Submit Preview", confirming the basic information of the voucher and entering the user payment password to complete the voucher registration operation. This voucher can be submitted for approval on the Pending Voucher page. Voucher batch registration is divided into two steps. The first step is to download the batch registration template, fill in the information related to the vouchers to be registered in an Excel template, which supports up to 50 vouchers in a single batch registration, and finish editing and uploading the Excel file. In the second step, a preview adjustment can be made to confirm the editing of the uploaded batch registration file. The bottom piece list line highlights any voucher that may be misidentified by the system and needs to be adjusted before confirming the registration. After confirmation, the voucher will enter the voucher to be submitted step. The design of the specific time sequence of the voucher registration operation is shown in Figure 4.

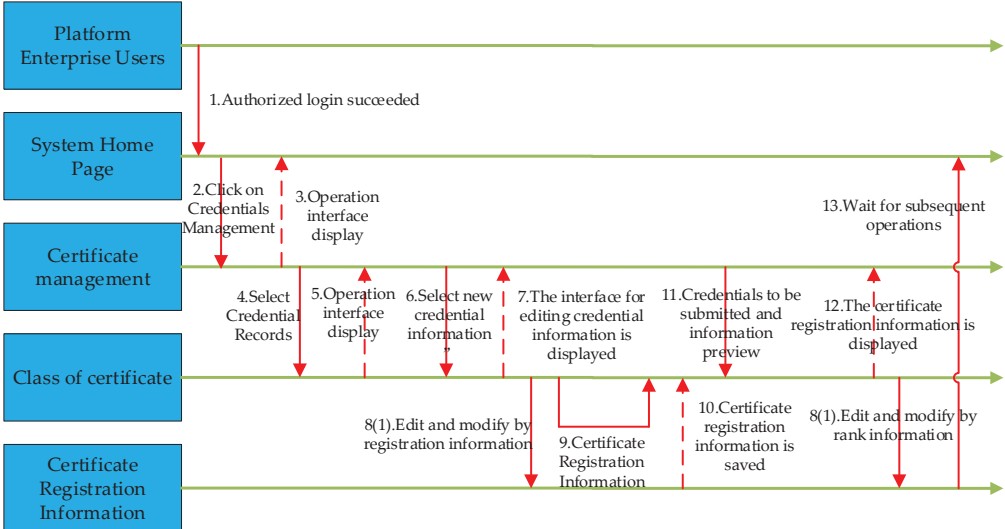

**Figure 4.** Schedule diagram of the certificate registration operation.

As seen from the timeline diagram of the voucher registration operation, after the platform enterprise user has been successfully authorized to enter the system homepage, they click on "Voucher Management" and then select "Voucher Registration" in the dis-

played interface. They then click "Add Voucher Information", and the user can edit and modify the voucher registration information in the displayed template or download the batch registration template. The voucher registration information is saved after the operation is completed, and then the user can submit the voucher and the subsequent voucher information preview and execute other related operations.

The voucher status management sub-module mainly includes two sub-functions: voucher status inquiry and voucher approval. The registered vouchers need to be submitted for approval on the voucher to be submitted page. When you perform voucher approval, you can see the basic information, the trade information, the payment commitment letter and the historical operation log information of the voucher. The voucher approval return or approval pass operation is performed based on the actual reference judgment. Once a voucher has been returned for approval, the applicant may resubmit it after modifying it on the returned voucher page. After the voucher is approved, the payment commitment letter needs to be signed and the payment password entered, then the voucher issuance operation is completed and the signing party can perform the sign-off operation. From the workflow diagram, it can be seen that after entering the "Voucher Status Management" operation interface, the operator can query the submitted vouchers and wait for the voucher approval. After approval, the operator can sign the payment commitment letter. After signing, they can directly issue the vouchers and finally wait for the vouchers to be signed.

The voucher receipt and transfer sub-module mainly includes three sub-functions: voucher receipt, transfer application and transfer approval. For vouchers issued by downstream core enterprises or vouchers transferred by downstream suppliers for sign-off operations, when a user performs voucher sign-off, they can see the base information of the voucher, trading company information, traceability information and payment commitment letter. The voucher signing or rejection operation can be executed after reference judgment according to the actual situation.

### 3.3.3. Detailed Design of Online Financing Module

Based on the system requirements analysis, we have learnt that the online financing module is mainly used to realize the receipt of suppliers' vouchers by combining its own needs, selecting the various funders that have settled on the platform, and financing applications, repayment and other operations. The entire online financing module includes three main subfunctions: financing application, financing approval and viewing and financing repayment.

The financing application sub-module includes two sub-functions: editing and submitting financing applications. In the process of submitting a financing application, the enterprise user can edit the financing application information in the template after selecting the corresponding financing application module. After editing according to the corresponding prompts, the user clicks "Submit" directly and waits for the approval of the relevant personnel of the platform and the funders.

The financing approval and view sub-module mainly includes two sub-functions: financing view and financing approval. Using this module, you can approve and view financing. There are two levels of financing approval: preliminary and final. After the initial review is approved, it then needs to be approved by the final review. The status changes to pending final review and after the final review approval is passed/returned, the status changes to approval passed/returned. If the initial review is rejected, it directly changes to approval returned, and suppliers are alerted. For the financing application whose repayment method is billing financing, the interest rate and financing period need to be entered during the preliminary examination, and can be directly entered in the final examination after the preliminary examination is completed. The release of funds will be triggered after the final examination is passed. See Figure 5 for the design of the specific working time sequence of financing approval and viewing operation. As seen from the entire time sequence diagram of the financing approval and viewing operation, after entering the home page of the system, the platform's fund users click on "Online

Financing" and select "Financing Approval and Viewing". They then enquire about their financing application information and then carry out the preliminary financing examination operation. After the operation is completed, they can enter the subsequent financing final examination operation.

The financing repayment sub-module mainly includes two sub-functions: financing repayment processing and financing repayment inquiry. Using this module platform enterprise users can process the repayment information of platform funders, and platform funders users can view all outstanding and closed financing.

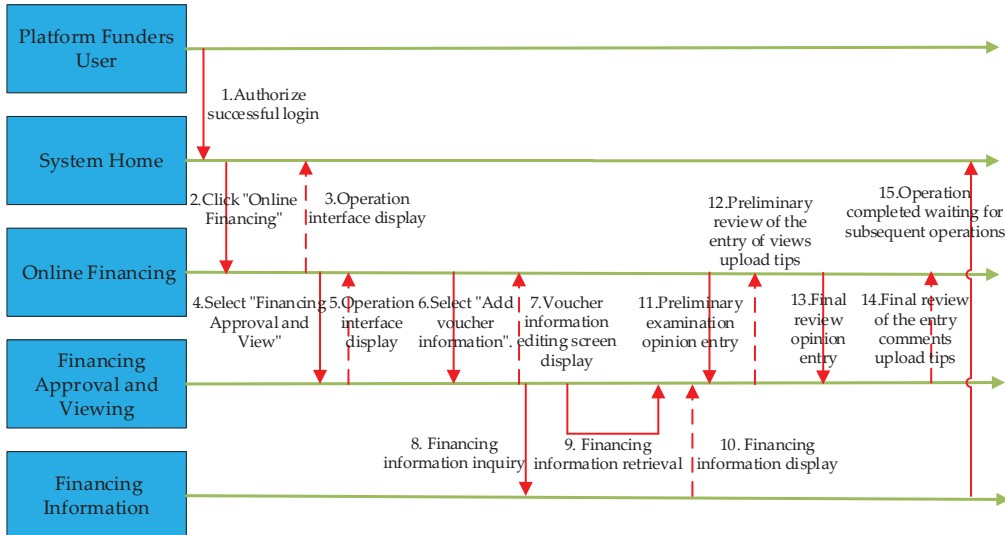

**Figure 5.** Financial approval and viewing operational schedule chart.

### 3.3.4. Detailed Design of Funds Management Module

As a result of what was learned from the system requirements analysis, the fund's management module is mainly used for the platform enterprise users to realize account management and top-up and withdrawal operations, for the enterprise funds users to realize automatic supervision of funds and funds flow and clearing operations. The fund management module mainly includes three sub-functions: account management, fund clearing and recharge and withdrawal.

The account management sub-module mainly includes two sub-functions: sub-account management and bank account management. After the name is verified, the sub-account will be automatically generated, and the platform enterprise users can view the basic information of the sub-account as well as the fund information and combine the sub-account information to bind and unbind the bank account. The added bank account name must be the same as the real name information.

The fund clearing sub-module mainly includes two sub-functions: automatic fund supervision and fund flow and clearing. By using blockchain to integrate into supply chain finance [37], the information is synchronized to the bank account payment system, and based on smart contracts, the automatic flow of funds, process supervision, and automatic settlement at maturity are realized. From the workflow diagram of fund clearing, it can be seen that the platform fund users enter the platform and click on "fund management", select "fund clearing", then select the required clearing content. The platform will generate the funds that meet the clearing conditions and the clearing data will be disseminated through the use of blockchain technology. Then, we judge whether the clearing Hash value is collected into the block, and if it has been collected into the block, workload proof is performed and full node verification is performed. This is followed by blockchain recording, and finally, the clearing information is synchronized to the bank account payment system [38].

The recharge and withdrawal sub-module mainly includes two sub-functions: recharge, and withdrawal. You must bind your bank card before recharging, otherwise, no recharge operation can be performed. A withdrawal fee is required, and the withdrawal can be withdrawn from any bank card. After filling in the withdrawal amount, the system will automatically calculate the corresponding handling fee, and the amount of the handling fee will be deducted from the sub-account after submitting the withdrawal.

*3.4. Conceptual Structure Design of the Database*

The system database design generally includes two aspects of database: logical structure design and data information table design. The logical structure of the database is generally designed using three common design methods: top-down, bottom-up and expansion-by-expansion [39]. The top-down approach defines and designs the overall data structure of the system from a macro perspective first, and then gradually refines the data structure from the top down until all the data of the system are defined. The bottom-up approach first designs the most basic data structure at the bottom of the system, and then gradually merges the basic data structures according to the logical relationships within the system, from the bottom up until the overall data structure of the system is obtained. The one-by-one expansion method first defines and designs the core data structure inside the system, and then gradually expands outward from the core data according to the internal logical relationship of the system, until the design of all data structures inside the system is complete [40].

ER diagrams, also known as entity-connection diagrams, provide a way to represent entity types, attributes and connections as a bridge between the real world and the program data world. Through ER diagrams, you can sort out the business relationships of the program, and ER diagrams are also a key issue in designing databases. In engineering, the database design and coding work can only be done if the business relationships are sorted out first. Based on the summarized requirements, the ER diagram has been established and the overall ER diagram of the whole financial management platform was designed as shown in Figure 6.

The data model is transformed into a compliant relational model by the above paradigm rules. For space reasons, the login registration information form will be selected for a detailed description of its format. The login registration information form includes username, invitation code, affiliated companies, login account, login password, confirmation password, date of registration and other content information. It is used to realize the entry and retrieval of login information for all new users in the financial management platform. The detailed design of the login information table is shown in Table 1.

**Table 1.** Transaction summary diagram.

| Field Name | Data Type | Length | Primary Key | Allow Empty | Description |
|---|---|---|---|---|---|
| certificate_id | varchar | 20 | yes | no | Voucher number |
| apfina_amount | decimal(16,4) | | | no | Amount of financing applied for |
| customer_name | varchar | 8 | | no | Customer name |
| certificate_amount | float | 4 | | no | Voucher amount |
| certificate_vouchers | float | 4 | | no | Voucher balance |
| compay_date | datetime | | | no | Committed payment date |
| original_registrar | varchar | 8 | | no | Original registered party |
| transferor | varchar | 8 | | no | Transferring party |
| application_time | datetime | | | no | No application time |
| trade_contracts | varchar | 20 | | no | Trade contracts |
| invoice | varchar | 8 | | no | Invoicing |
| certifinapurpose | varchar | 20 | | no | Proof of use of financing |

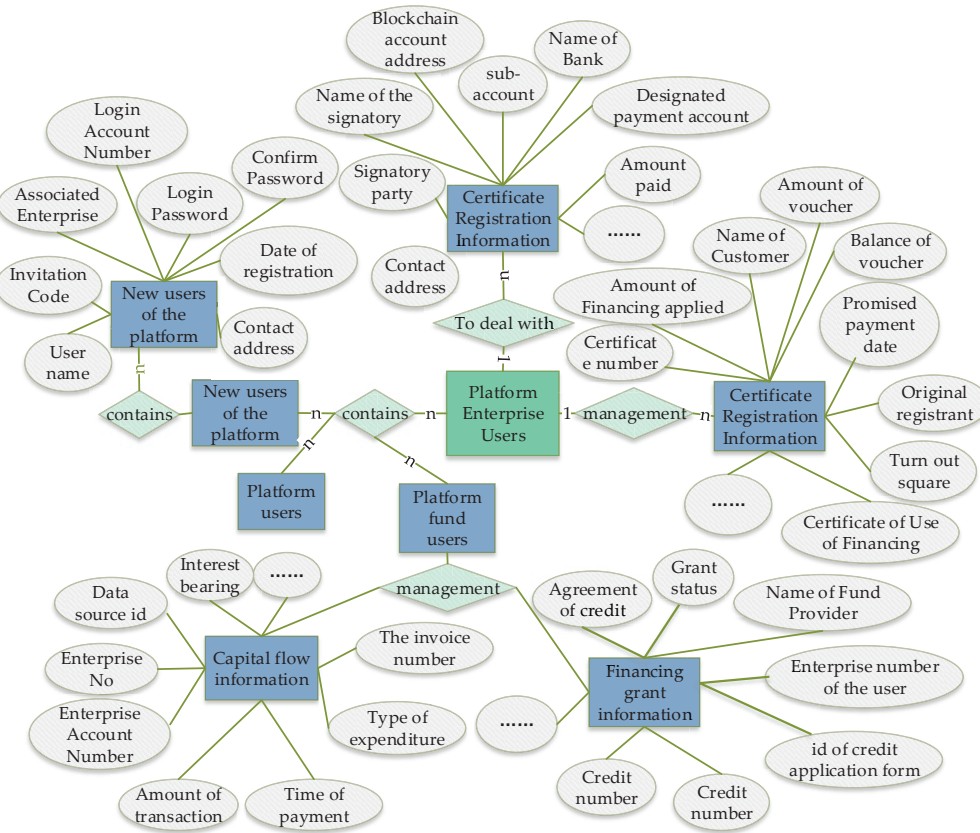

**Figure 6.** System ER diagram.

## 4. System Implementation

Through field research, we know that the platform developed by this topic needs to be multi-user oriented in practical application and needs to have the ability to handle a large number of users who can access simultaneously. When multiple users accessed the system at the same time and were able to perform normal services, the system response time could be controlled to within two seconds, as the amount of information accessed is small. When the amount of information accessed was large and reached a peak, the system response time could be controlled to within four seconds. When the system was busy, the system should have the corresponding operation prompt function, so that users can perform operations in a timely manner. Combined with the actual demand of the platform, the concurrent users of the whole platform are 300 people.

The system implementation was based on the system design and functional requirements analysis which described the specific details of the system implementation, generally including the implementation of system functional modules and the implementation of system interfaces. This section will describe the implementation of the main modules of the system and the actual interface implementation according to the overall design of the financial management platform.

The detailed design of the system describes the core functional modules, which mainly includes four modules: the login registration and account opening module, the voucher management module, the online financing module and the fund management module. When a user initiates a specific request to access the system, they first need to complete the authentication their identity through the system authentication system to obtain the relevant rights to access the system operations.

### 4.1. Login Registration and Account Opening Module

From the detailed design content of the financial management platform, it can be confirmed that the entire login registration and account opening module mainly includes

three sub-functions: login registration, real name authentication and sub-account opening. The specific login registration and account opening class diagram design is shown in Figure 7.

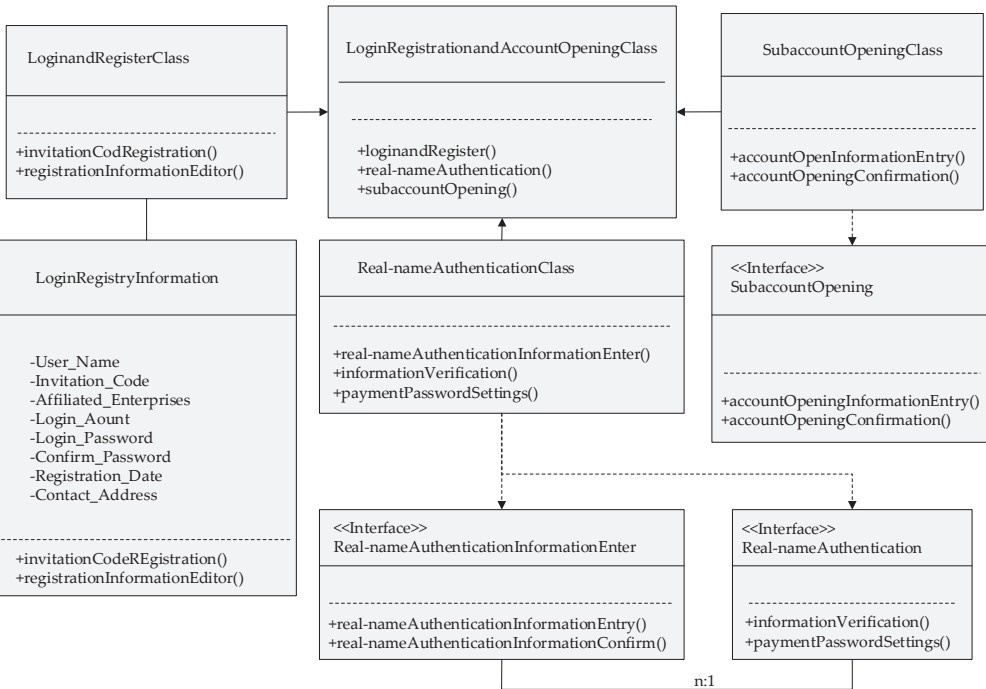

**Figure 7.** Class diagram of login registration and account opening modules.

According to the functional design and detailed design of the login registration and account opening module, in the process of invitation code registration, users need to edit and enter the invitation code, the associated companies, the login account number, the login password, the confirmation password and other details. The user then clicks "Confirm" directly after the editing is completed. In the process of a real name authentication operation, the user needs to fill in the company name of the key used and the unified social credit certificate code (hereinafter referred to as credit code), insert the certificate DN code and then click on the confirmation button to complete. In the process of real name authentication, the company name, the credit code and the certificate are all the same if under the same business.

### 4.2. Voucher Management Module

According to the detailed design of the financial management platform, the whole voucher management module mainly includes three sub-functions of voucher registration, voucher status management and voucher sign-in and transfer. The design of the specific voucher management class diagram is shown in Figure 8.

According to the detailed design of the voucher management module, the voucher registration can be done by inviting upstream suppliers with real names to issue vouchers, fill in the signatory and payment-related information and choose whether to upload accompanying trade information. The suppliers then click "Submit Preview", confirm the basic information of the voucher, and enter the user payment password to complete the voucher registration operation. When performing voucher approval, the user can see the voucher's base information, trade information, payment commitment letter and historical operation log information. The voucher approval return or approval pass operation is performed based on the actual reference judgment. Once a voucher has been returned for approval, the applicant may resubmit it after modifying it on the returned voucher page. After the voucher is approved, the payment commitment letter needs to be signed and the payment

password is entered, then the voucher issuance operation is completed and the signing party can perform the sign-off operation.

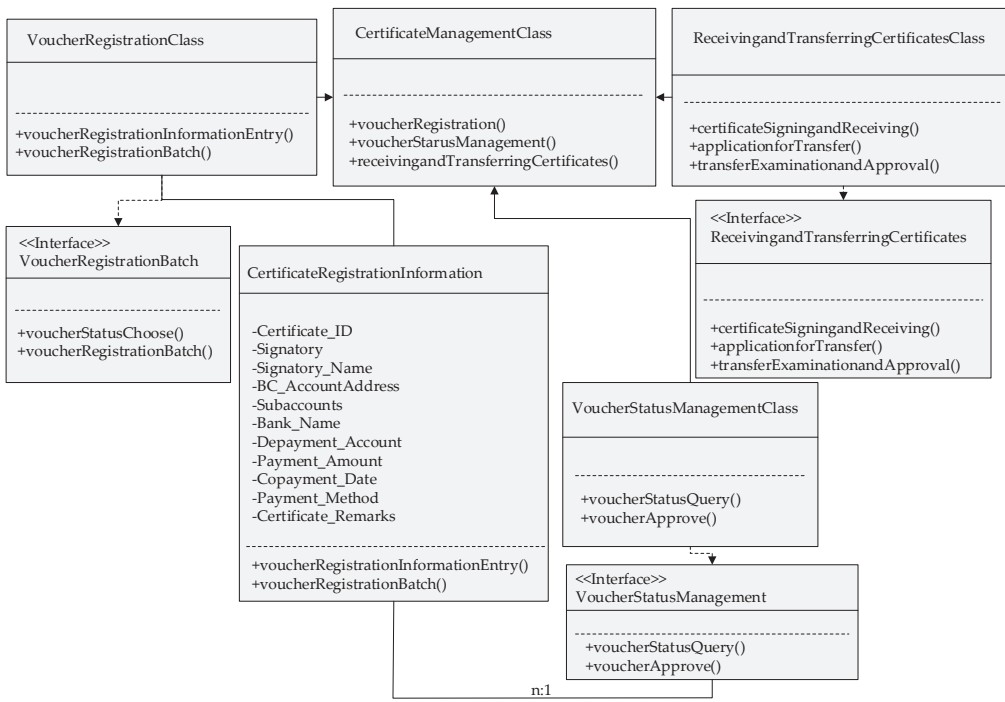

**Figure 8.** Class diagram of certificate management module.

*4.3. Online Financing Module*

From the detailed design of the financial management platform, the entire online financing module mainly includes three major sub-functions: financing application, financing approval and view and financing repayment. The specific design of the online financing module class diagram is shown in Figure 9.

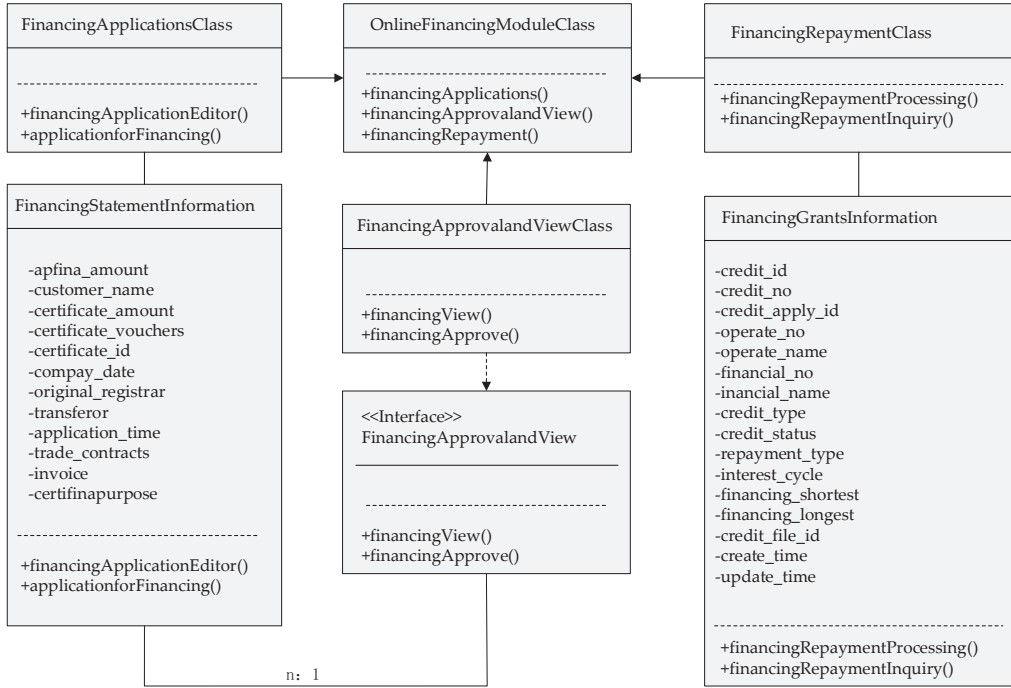

**Figure 9.** Class diagram of online financing modules.

Based on the detailed design of the online financing module. In the process of financing approval operation, for the preliminary examination of financing, for financing applications whose repayment method is billing financing, the interest rate and financing period need to be entered in the preliminary examination. The system also displays information on the interest rate, operation time, approval opinion and supporting documents. In the process of a financing repayment inquiry, the use can view all outstanding and closed financing from the specific implementation operation web pages. In addition, the user can also check specific repayment records and financing details, such as the financing number, the financing amount, the interest rate and the repayment method and status.

### 4.4. Funds Management Module

According to the detailed design of the financial management platform, the whole fund management module mainly includes three major sub-functions of account management, fund clearing and top-up and withdrawal. The specific operation can be found in Figure 10.

Based on the detailed design of the fund's management module, after completing the real name, the corresponding sub-account will be automatically created, and then the user can view the basic information and fund information of the sub-sub-account, as well as top up and withdraw funds to the sub-account. The billing history will show all the transactions; this page only shows the latest 10 entries. The use can click "All Bills" to view all the billing information, where billing information mainly includes basic information such as outgoing/incoming time, line number, voucher number, type, status, incomings, outgoings and balance. The user must link their bank card before recharging, otherwise they cannot perform the recharge operation. The user enters the recharge amount, enters the payment password of the set system, and clicks on "confirm recharge" to submit the application.

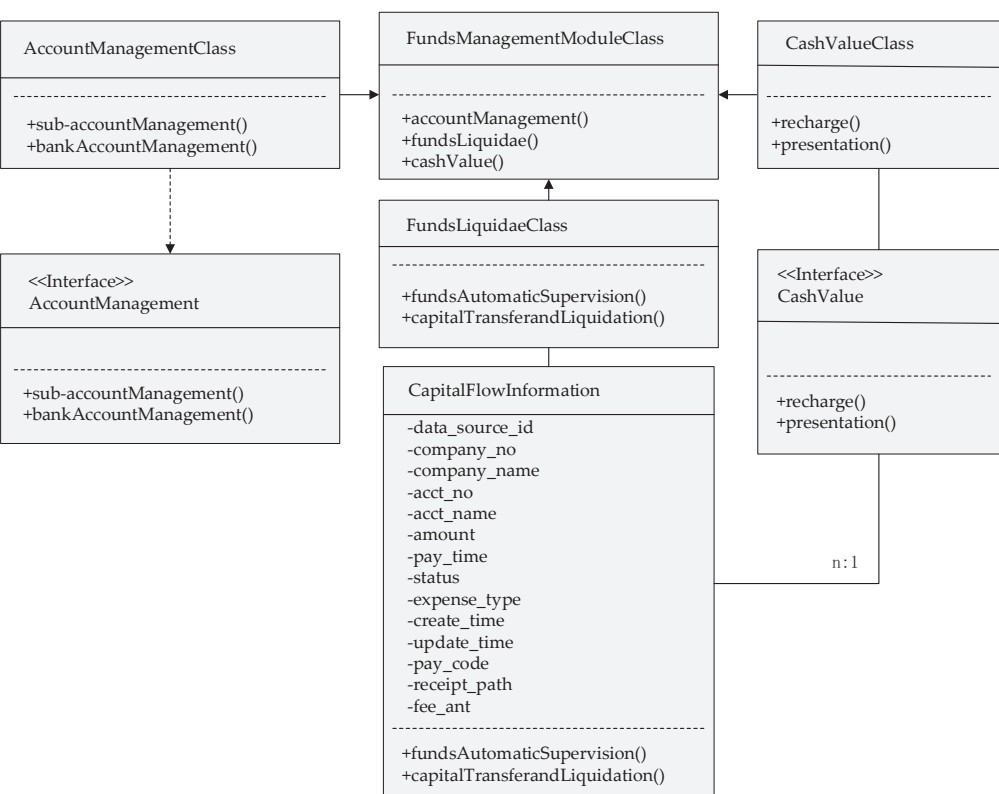

**Figure 10.** Funds management class diagram.

## 5. Test

After the development of the system was completed, in order to ensure that the system functions properly and the business processes and performance meet the design requirements, it was necessary to conduct a comprehensive scientific test of the system to find various defects and bugs in the system and make timely improvements.

### 5.1. System Functionality Testing

This section will test the implementation of the whole financial management platform. Due to the limited space, only the project login and account opening module and online financing module were used in the case-specific test situation described here. The specific functional testing of the login and account opening module is shown in Table 2. From the whole functional testing table of the login and account opening module, it can be seen that in the process of functional testing of this module, the two sub-modules of the login and account opening module, login registration and sub-account opening were tested comprehensively.

Specific functional tests of the online financing module are shown in Table 3. As can be seen from the entire online financing module function test table, in the process of this module function test, mainly the financing application editing and financing application submission in the financing application module and the financing approval and view sub-module in the financing view and financing approval related operations were tested comprehensively, and the test results were able to meet the design requirements and index regulations.

**Table 2.** Login registration and account opening function test case list.

| Serial Number | Purpose of Test | Prerequisites | Specific Operations | Expected Results | Test Results |
|---|---|---|---|---|---|
| GNCS-01 | Implementing new user login and registration operations. | Users receive the registration invitation code, click "Register" and successfully enter the registration interface. | Click on "Register" in the platform, and edit the information in the registration screen and save it. | The user completes the editing of the registration information. The platform can prompt "Registration successful". | pass |
| GNCS-02 | Realize subaccount opening operation. | After successful registration, new users will be able to enter the subaccount opening operation interface at the system prompt. | Click on "Open an Account" in the platform and edit the information in the account opening operation interface. | The new user completes the editing of the account opening information and clicks Save to open the sub-account. | pass |

**Table 3.** Online financing module function test case list.

| Serial Number | Purpose of Test | Prerequisites | Specific Operations | Expected Results | Test Results |
|---|---|---|---|---|---|
| GNCS-03 | Realize financing application operation. | The user successfully enters the financing application operation screen. | Click "Online Financing" in the platform, select "Financing Application", and edit the information of financing application. | Users are able to edit, save and submit financing application information, and check its status. | pass |
| GNCS-04 | Realize financing approval operation. | The user successfully enters the financing approval operation interface. | In the platform, click "Online Financing" and select "View Financing Application" to check and approve new financing applications. | Users are able to query the details of new financing applications and to enter and submit preliminary and final comments in conjunction with the actual. | pass |

### 5.2. System Performance Testing

System performance testing is the tests that the system is running, including a stress test, a load test, and so on. In the test of the financial management platform, the Load Runner Tool stress test tool was mainly used. After starting the Load Runner Tool, the function simulates user access and conducts stress tests. In the process of stress testing, combined with the system non-functional requirements, it can be seen that the whole platform can handle 300 concurrent users, with a the duration of about 6 h. The specific platform concurrent user performance test statistics are shown in Table 4 and the test transaction summary is shown in Table 5.

**Table 4.** Summary table of statistics.

| Content | Result |
| --- | --- |
| Summary table of statistics | 300 |
| Total throughput (bytes) | $1.8088475899 \times 10^{11}$ |
| Average throughput (bytes per second) | 8,361,520 |
| Total number of clicks | 7,934,380 |
| Average number of clicks per second | 366.722 |
| Total number of errors | 3926 |

**Table 5.** Transaction summary diagram.

| Name of Transaction | Minimum Value | Mean Value | Maximum Value | Standard Deviation | 90 Percent | Success | Failure | Stop |
| --- | --- | --- | --- | --- | --- | --- | --- | --- |
| Action Transaction | 0.131 | 4.807 | 114.154 | 1.008 | 6.051 | 1,324,818 | 3926 | 0 |
| User End Transaction | 0 | 0 | 0 | 0 | 0 | 300 | 0 | 0 |
| User Initiate Transaction | 0 | 0 | 0.002 | 0.001 | 0.002 | 300 | 0 | 0 |
| Home Page Interface | 0.131 | 4.807 | 114.154 | 1.008 | 6.051 | 1,324,818 | 3926 | 0 |

The detailed concurrent user performance tests are tabulated in Table 6. Combined with the whole test situation table, it can be seen that when the number of test users gradually rises from 50 to 300, the platform ensures normal operation and the response time of the whole platform can be guaranteed within five seconds, fully meeting the performance requirements.

**Table 6.** Concurrent user performance test sheet.

| Number of Test Users | Expected Test Value (seconds) | Actual Test Value (seconds) |
| --- | --- | --- |
| 50 | 2.0 | 1.657 |
| 100 | 3.0 | 2.934 |
| 200 | 4.0 | 3.778 |
| 300 | 5.0 | 4.832 |

The specific transaction summary statistics test results are shown in Figure 11. Combined with the specific situation of the transaction summary statistics, it can be seen through the HTTP response results that in the absence of 404 and 500 errors, transaction failures accounted for about 0.3 percent of the total transactions, which was within the normal indicators, and therefore fully meet the design index requirements.

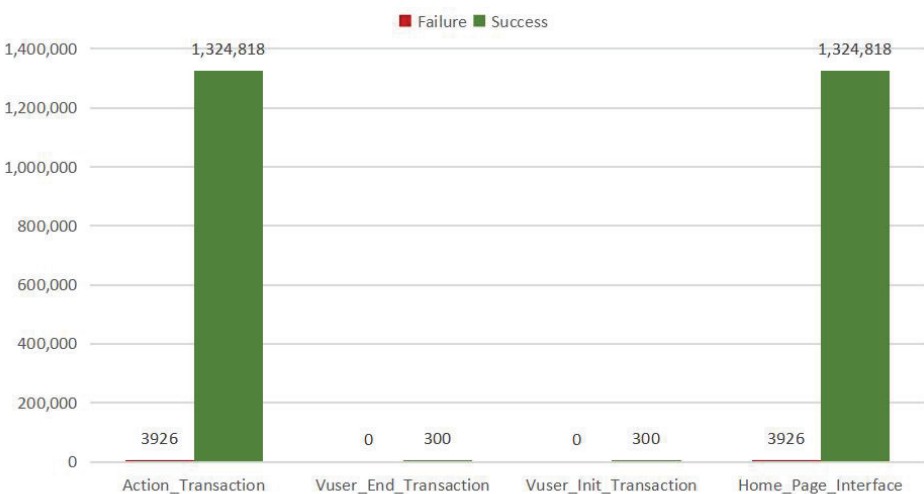

**Figure 11.** Transaction summary statistical test situation diagram.

## 6. Conclusions

In this paper, we have designed and developed a financial platform combining the blockchain and supply chain, based on the analysis of the business needs and system functional requirements of each participant in the supply chain model. The system provides good support for the daily financial management of enterprises, and is of great significance for improving financing efficiency and increasing capital flow. The financial management platform leverages the characteristics of blockchain to improve the processing process of transactions and the security of the system. For core enterprises, after accessing the supply chain financial platform, they can quickly meet the funding needs of multi-level suppliers with their good credit qualifications, and carry out financing allocation to solve the financing problems of small- and medium-sized enterprises in the business chain. After the development platform is put into use, it can further facilitate subsequent financing, loans and other financial services. In addition, using data mining, decision support and other data analysis technologies to process the stored supply chain data can provide support for risk assessment and decision-making, and help establish effective credit rating models and risk management mechanisms based on the analysis results.

**Author Contributions:** Conceptualization, H.L. and B.Y.; methodology, S.Z.; software, B.C.; validation, X.X., A.T. and X.Z.; formal analysis, H.L.; investigation, B.Y.; resources, A.T.; data curation, H.L.; writing—original draft preparation, H.L.; writing—review and editing, X.X.; visualization, B.Y.; supervision, S.Z.; project administration, X.Z.; funding acquisition, A.T. All authors have read and agreed to the published version of the manuscript.

**Funding:** This work was funded by the Researchers Supporting Project number (RSPD2023R681) King Saud University, Riyadh, Saudi Arabia.

**Institutional Review Board Statement:** Not applicable.

**Informed Consent Statement:** Not applicable.

**Data Availability Statement:** Not applicable.

**Conflicts of Interest:** The authors declare no conflict of interest.

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
