# Peer review of "A Financial Management Platform Based on the Integration of Blockchain and Supply Chain"

_sensors, doi:10.3390/s23031497_

Round 1

Reviewer 1 Report

In this paper, the authors study the security issues in the financial management platform and design a financial management platform by integrating the features of blockchain technology and supply chain technology. This paper applies the characteristics of blockchain technology to solve the problems that the credit of core enterprises in the supply chain finance industry cannot be transmitted, the business information of the whole chain cannot be connected, and the clearing and settlement are cumbersome. The theoretical derivation and performance analysis illustrate the effectiveness of the proposed solution.In general, the problems studied are important and the solutions proposed are attractive. I have the following comments: 

1.The context of the article is the financial industry, which has high requirements for data security, how did the author consider this aspect?

2.It is recommended that a more detailed summary of the blockchain work be provided in Section 2 Related Work.

3.In order to facilitate mastering the overall system architecture before delving into the details of the system, a better description of the overall framework is necessary in Figure 2.

4.The requirements are introduced in more detail in section 3.1, but the topic blockchain is introduced relatively little, and it is recommended to add the blockchain content appropriately.

5.The database design is under-described in Section 3.4, and the authors need to elaborate further on this section.

Author Response

Comment 1: The context of the article is the financial industry, which has high requirements for data security, how did the author consider this aspect?

Response 1: This paper reflects data security in non-functional requirements. This paper provides a detailed description of the data storage security aspects, as well as prevention of possible problems. The rest of this paper will focus on the improvement of the performance of each function of the system.

Comment 2: It is recommended that a more detailed summary of the blockchain work be provided in Section 2 Related Work.

Response 2: Thanks for reviewer’s comment. We have added references [26] and [27] to the related work in the revised version of the manuscript, which presents more clearly the advantages offered by the combination of blockchain technology with other technologies.

Comment 3: In order to facilitate mastering the overall system architecture before delving into the details of the system, a better description of the overall framework is necessary in Figure 2.

Response 3: Thanks for reviewer’s comment. We have added a description of Figure 2 in Section 3.2 of the revised version of our manuscript to convey the meaning of the picture. We have revised in Section 3.2 part of the manuscript as: “The user display interface is used to show the core business processes, and the business support is used to ensure the normal operation of the system backend. The display layer and the business processing layer form the entire system framework. The data layer is used to ensure the security of data and the recall of system data. “ in lines 193-197.

Comment 4: The requirements are introduced in more detail in section 3.1, but the topic blockchain is introduced relatively little, and it is recommended to add the blockchain content appropriately.

Response 4: Thanks for reviewer’s comment. We have revised in Section 3.1 part of the manuscript as:”Funds management is the core function of the whole financial management platform, which is the integration of blockchain technology and supply chain finance technology. It relies on the decentralized, tamper-evident and smart contract features of blockchain [33], synchronizes the bank account payment system, and realizes the automatic flow of funds, process supervision and automatic settlement at maturity. “ in lines 141-146.

Comment 5: The database design is under-described in Section 3.4, and the authors need to elaborate further on this section.

Response 5: Thanks for reviewer’s comment. we have revised in Section 3.4 part of the manuscript as: ”The data model is transformed into a compliant relational model by the above paradigm rules.For space reasons, the login registration information form will be selected for a detailed description of its format. The Login registration information form includes the user name, invitation code, affiliated companies, login account, login password, confirmation password, date of registration and other content information. It is used to realize the entry and retrieval of login information for all new users in the financial management platform, etc. The detailed design of the login information table is shown in Table 1.” in lines 416-422. And we also added Table 1.

Thanks again for your insightful corrections and time spent on this paper.

Reviewer 2 Report

The security issues in the financial management platform are studied in this paper. The authors solve the problems from the perspective of blockchain to handle the problem that the credit of core enterprises in supply chain finance cannot be transmitted and the whole chain business cannot be connected. The effectiveness and rationality of the proposed financial management platform is verified through rigorous experiments. Overall, the paper was interesting and well organized. I have some comments to further improve the quality of this paper:

1.In Section 3.1, the reliability of the system is judged based on the probability that whether the system fails within a specified period of continuous operation being less than a preset value. How to set this value is not clear?

2. Why is it necessary to choose MVC architecture and what are its advantages over other architectures?

3. In section 3.2, the role of the presentation layer, business processing layer and data layer in this system was explained. What are the relationship between them? More details are required.

4.Section 3.3 introduces the core business part of the system architecture, and section 3.4 introduces the data storage part of the system architecture, but there is a business support part in the overall architecture of your system that is not reflected in the text. Thus, the authors should provide more explanations for the business support part.

5. More related works reflecting new trends in IOT should be analyzed, i.e. "Edge Computing for Internet of Everything: A Survey." IEEE Internet of Things Journal 9, no.(23), pp.23472–23485, 2022.

Author Response

Comment 1: In Section 3.1, the reliability of the system is judged based on the probability that whether the system fails within a specified period of continuous operation being less than a preset value. How to set this value is not clear?

Response 1: Thanks for reviewer’s comment. We have revised in Section4 part of the manuscript as: “Through field research, we know that the platform developed by this topic needs to be multi-user oriented in practical application, and needs to have the ability to handle a large number of users in parallel access. When multiple users access the system at the same time and are able to perform normal services, the system response time can be controlled within two seconds when the amount of information accessed is small. When the amount of information accessed is large and the amount of information reached a peak, the system response time can be controlled within four seconds. When the system is busy, the system should have the corresponding operation prompt function, so that users can perform operations in a timely manner. Combined with the actual demand of the platform, the concurrent users of the whole platform are 300 people.” in line 424-433. The presets are set in Table5.

Comment 2: Why is it necessary to choose MVC architecture and what are its advantages over other architectures?

Response 2: MVC architecture adopts the hierarchical design concept, greatly reducing the degree of coupling between layers, making the platform architecture more clear, and effectively improving the scalability and maintenance of the platform. After the system business changes, it is not necessary to directly update all the structure of the system, but only need to extend and maintain the business layer according to the business requirements, thus greatly improving the maintainability of the system. On the other hand, MVC hierarchical architecture greatly improves the availability of code, which plays a positive role in improving the efficiency of system development and reducing the cost of system development. Through the above analysis and comparison, in order to ensure strong scalability and availability of the whole platform, MVC architecture is finally selected as the design architecture of the whole platform.

The advantages of MVC architecture over other architectures: each performs its own duties without interfering with each other; In the MVC pattern, each of the three layers does its job, so if the requirements change in one layer, the code in that layer needs to be changed without affecting the code in the other layers.

Comment 3: In section 3.2, the role of the presentation layer, business processing layer and data layer in this system was explained. What are the relationship between them? More details are required.

Response 3: Thanks for reviewer’s comment. We have revised in Section3.2 part of the manuscript as: “MVC hierarchical architecture can easily complete distributed deployment: The bottom layer is the presentation layer, which is mainly responsible for the display of the interface, directly facing the user; The second layer is the application system service layer, which is usually deployed in one or more servers and is mainly responsible for completing the logical processing of all kinds of data in the system; The third layer is the data layer, which is mainly composed of the database system and mainly responsible for the management of all kinds of data in the system.” in line 215-221.

Comment 4: Section 3.3 introduces the core business part of the system architecture, and section 3.4 introduces the data storage part of the system architecture, but there is a business support part in the overall architecture of your system that is not reflected in the text. Thus, the authors should provide more explanations for the business support part.

Response 4: Thanks for reviewer’s comment. Section 4 of the article introduces the implementation method of the system in detail, and this section provides a relatively detailed explanation of the business support part. For example, the login registration and account opening module mainly includes three sub-functions: login registration, real name authentication, and sub-account opening; the voucher management module includes three sub-functions of voucher registration, voucher status management, and voucher sign-in and transfer; the online financing module mainly includes three major sub-functions: financing application, financing approval, and view, and financing repayment and the fund management module includes three major sub-functions: account management, fund clearing, and top-up and withdrawal.

Comment 5: More related works reflecting new trends in IOT should be analyzed, i.e. "Edge Computing for Internet of Everything: A Survey." IEEE Internet of Things Journal 9, no.(23), pp.23472–23485, 2022.

Response 5: Thanks for reviewer’s comment. We analyzed more related work reflecting the new trend of the Internet of things, such as "Edge Computing for Internet of Everything: A Survey." IEEE Internet of Things Journal 9, no.(23), pp.23472 -- 23485, 2022. and "Dynamic UAV Deployment for Differentiated Services: A Multi-Agent Imitation Learning Based Approach," IEEE Transactions on Mobile Computing, Doi: 10.1109/TMC.2021.3116236, 2021. We have added these two references in the revised version of the manuscript, namely reference [11] and [15].

Round 2

Reviewer 1 Report

The authors have addressed all my concerns. No further comments.